# The crystal structure of Staufen1 in complex with a physiological RNA sheds light on substrate selectivity

Daniela Lazzaretti[1,]*, Lina Bandholz-Cajamarca[1,]*, Christiane Emmerich[1], Kristina Schaaf[1], Claire Basquin[2], Uwe Irion[1], Fulvia Bono[1,3]

**During mRNA localization, RNA-binding proteins interact with specific structured mRNA localization motifs. Although several such motifs have been identified, we have limited structural information on how these interact with RNA-binding proteins. Staufen proteins bind structured mRNA motifs through dsRNA-binding domains (dsRBD) and are involved in mRNA localization in *Drosophila* and mammals. We solved the structure of two dsRBDs of human Staufen1 in complex with a physiological dsRNA sequence. We identified interactions between the dsRBDs and the RNA sugar–phosphate backbone and direct contacts of conserved Staufen residues to RNA bases. Mutating residues mediating nonspecific backbone interactions only affected Staufen function in *Drosophila* when in vitro binding was severely reduced. Conversely, residues involved in base-directed interactions were required in vivo even when they minimally affected in vitro binding. Our work revealed that Staufen can read sequence features in the minor groove of dsRNA and suggests that these influence target selection in vivo.**

## Introduction

The intracellular localization of mRNAs is a critical gene regulatory mechanism in many eukaryotic cell types and processes (reviewed in Martin & Ephrussi [2009]; Meignin & Davis [2010]; Medioni et al [2012]; Chin & Lécuyer [2017]). It targets the synthesis of proteins to their site of function in the cytoplasm, thus providing a fine spatial control of gene expression. A key step in the pathway is the specific recognition of the localizing transcript, which depends on linear or structured *cis*-acting signals, generally located in the mRNA 3′UTR. These *cis* elements are bound by *trans*-acting factors, or RNA-binding proteins that mediate the recruitment of additional components required for transport, anchoring, and translational control of the mRNA. One of the best-characterized *trans*-acting factors, with many functions in RNA metabolism, is the protein Staufen (Stau).

Stau was first identified in *Drosophila melanogaster* as a critical factor for mRNA localization. During oogenesis, *Drosophila* Stau (dStau) is required for *oskar* (*osk*) mRNA transport to the posterior pole of the oocyte, where abdomen and germ cells will develop (Ephrussi et al, 1991; Kim-Ha et al, 1991, 1995, St Johnston et al, 1991, 1992). In the mature egg, dStau accumulates at the anterior pole and anchors *bicoid* (*bcd*) mRNA, the anterior determinant (St Johnston et al, 1989; Ferrandon et al, 1994). During the asymmetric division of embryonic neuroblasts, dStau associates with *prospero* (*pros*) mRNA and restricts its localization to the daughter cell, which will differentiate into a ganglion mother cell (Broadus & Doe, 1997; Li et al, 1997; Broadus et al, 1998; Fuerstenberg et al, 1998; Matsuzaki et al, 1998; Schuldt et al, 1998; Shen et al, 1998).

Stau homologs have been described in numerous organisms, including mammals (Buchner et al, 1999; Marión et al, 1999; Wickham et al, 1999), *Xenopus laevis* (Yoon & Mowry, 2004), *Danio rerio* (Bateman et al, 2004), *Aplysia californica* (Liu et al, 2006), and *Caenorhabditis elegans* (LeGendre et al, 2013). In mammals, two Stau homologs have been identified: Stau1, which is ubiquitously expressed, and Stau2, enriched in the brain (Wickham et al, 1999; Tang et al, 2001).

Apart from their conserved role in mRNA localization (reviewed in Heraud-Farlow & Kiebler [2014]), Stau proteins act in the control of mRNA stability (Kim et al, 2005, 2007; Heraud-Farlow et al, 2013; Kretz et al, 2013; Park et al, 2013), translation (Micklem et al, 2000; Dugré-Brisson et al, 2005), splicing (Ravel-Chapuis et al, 2012), and nuclear export (Elbarbary et al, 2013). In addition, Stau proteins have been implicated in RNA interference (LeGendre et al, 2013) and have been shown to be important for the replication of RNA viruses, such as HIV and influenza virus (Falcón et al, 1999; Mouland et al, 2000; Chatel-Chaix et al, 2004; de Lucas et al, 2010).

Members of the Stau protein family share a common domain organization, consisting of four to five copies of the dsRNA-binding domain (dsRBD) separated by linkers that are predicted to be unstructured (St Johnston et al, 1992; Marión et al, 1999; Wickham

---

[1]Max Planck Institute for Developmental Biology, Tübingen, Germany   [2]Max Planck Institute of Biochemistry, Martinsried, Germany   [3]Living Systems Institute, University of Exeter, Exeter, UK

Correspondence: f.bono@exeter.ac.uk
*Daniela Lazzaretti and Lina Bandholz-Cajamarca contributed equally to this work.

et al, 1999; Micklem et al, 2000; LeGendre et al, 2013) (Fig 1A). Mammalian Stau proteins also contain a domain capable of binding tubulin in vitro (Tubulin-Binding Domain; Wickham et al, 1999), and a motif involved in Stau homo-dimerization (Stau-swapping motif [SSM]; Gleghorn et al, 2013).

Interestingly, each Stau dsRBD is more similar to the corresponding domain in a different Stau homolog than it is to another dsRBD within the same protein (Micklem et al, 2000) (Fig S1A and B). Consistent with the sequence divergence, Stau dsRBDs have distinct functionalities: only dsRBD1, dsRBD3, and dsRBD4 bind dsRNA in vitro, whereas dsRBD2 and dsRBD5 lack RNA-binding activity and likely act as protein–protein interaction domains (St Johnston et al, 1992; Wickham et al, 1999; Micklem et al, 2000; Ramos et al, 2000). Indeed, dStau dsRBD5 is important for translational activation of osk mRNA (Micklem et al, 2000) and also interacts with the actin-binding protein Miranda (Mira) to localize pros mRNA (Fuerstenberg et al, 1998; Schuldt et al, 1998; Shen et al, 1998; Irion et al, 2006; Jia et al, 2015); in hStau1, dsRBD5, together with the SSM, mediates homo-dimerization (Martel et al, 2010; Gleghorn et al, 2013). A long insertion within dStau dsRBD2 (Fig S1A) is instead required for microtubule-dependent localization of osk mRNA (Micklem et al,

2000). In hStau1, dsRBD4 and TDB recruit the nonsense-mediated decay factor Upf1 to trigger Stau-mediated mRNA decay (SMD) (Kim et al, 2005).

The dsRBD is a small domain, of 65–70 aa, present in a variety of proteins involved in almost every aspect of RNA metabolism (reviewed in Masliah et al [2013]; Tian et al [2004]). The domain binds RNA mainly through electrostatic interactions with the RNA backbone, and selects A-form RNA double helices by shape complementarity, establishing contacts with two subsequent minor grooves and the intervening major groove of the dsRNA. Thus, the binding of a dsRBD to dsRNA has generally been considered to be dependent on the target RNA structure, rather than sequence. Human, Drosophila, and C. elegans Stau were in fact shown to bind dsRNA without apparent sequence specificity in vitro (St Johnston et al, 1992; Marión et al, 1999; Wickham et al, 1999; Ramos et al, 2000; LeGendre et al, 2013; Wang et al, 2015). Yet, in all these systems, Stau proteins associate with specific RNA targets (St Johnston et al, 1992; Ferrandon et al, 1994; Li et al, 1997; Mallardo et al, 2003; Kim et al, 2005, 2007; Furic et al, 2008; Heraud-Farlow et al, 2013; Laver et al, 2013; LeGendre et al, 2013; de Lucas et al, 2014; Ricci et al, 2014;

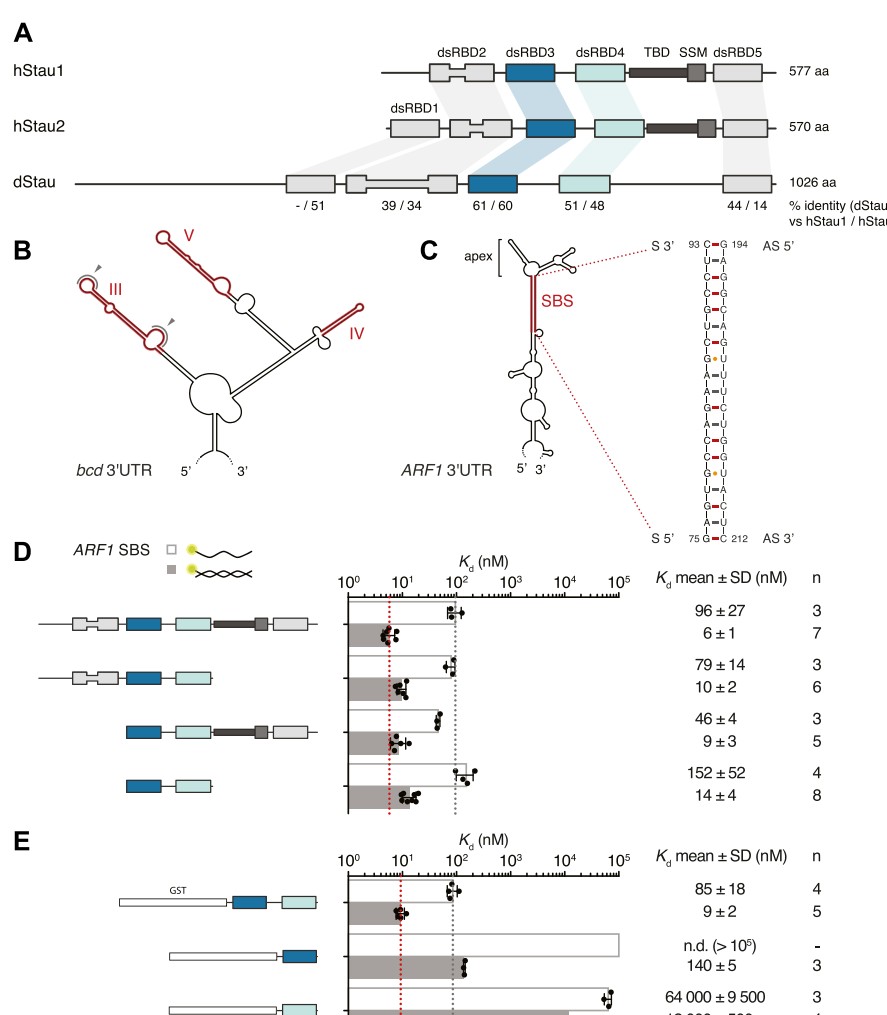

**Figure 1. dsRBDs 3 and 4 of hStau1 are sufficient for ARF1 SBS binding.**
**(A)** Schematic representation of the domain architecture of human (h) and Drosophila (d) Stau. Lines indicate regions predicted to be unstructured; boxes represent folded domains. The insertion in dsRBD2 is indicated by thinning of the box. Percentage identity between each dsRBD of dStau compared with hStau1 or hStau2 is indicated. **(B)** Secondary structure representation of the central part of bcd 3′UTR (as in Brunel & Ehresmann [2004]). The distal regions of stem-loops III, IV, and V, required for dStau binding, are highlighted in red. Loops within region III that mediate bcd mRNA homo-dimerization are indicated by gray arrows. **(C)** Predicted secondary structure of ARF1 3′UTR (as in Kim et al [2007]). The 19 bp stem structure required for hStau1 binding (SBS, Stau-Binding Site) is highlighted in red; its sequence is shown on the right. **(D, E)** $K_d$ values determined by FA, using 5′-fluorescein-labeled ARF1 SBS, either single (sense strand) or double stranded, and the recombinantly purified hStau1 constructs indicated in the schematics. The graphs show mean $K_d$ (bars), standard deviation (black lines), and $K_d$ values obtained in each independent experiment (black dots). Mean $K_d$ ± SD, in nM, and number of independent measurements (n) are indicated on the right. AS, antisense strand; S, sense strand; SSM, Stau-swapping motif; TBD, tubulin-binding domain.

Sugimoto et al, 2015), raising the question of how these targets are recognized in vivo.

Several high-throughput studies have sought to define the RNA elements recognized by Stau proteins: identified Stau-binding sites (SBS) reside in double-stranded regions, formed either intra- or intermolecularly, have an average length of 9–12 bp, with none or few mismatches, but are otherwise poorly defined (Heraud-Farlow et al, 2013; Laver et al, 2013; LeGendre et al, 2013; de Lucas et al, 2014; Ricci et al, 2014; Sugimoto et al, 2015). For a few targets, Stau-dependent *cis*-acting elements have been mapped and validated in vivo (Fig 1B and C). The localization of *bcd* mRNA requires the structural integrity of stem-loop regions III, IV, and V in its 3′UTR (Ferrandon et al, 1994, 1997), in addition to intermolecular homo-dimerization of the mRNA mediated by loop III (Ferrandon et al, 1997). Similarly, a stem of 19 bp is required for the binding of human Stau1 (hStau1) to the 3′UTR of *ADP-Ribosylation Factor 1* (*ARF1*) mRNA. *ARF1* mRNA bound to hStau1 is targeted for degradation through SMD (Kim et al, 2005, 2007; Park & Maquat, 2013).

To understand how Stau recognizes its target RNAs, we determined the crystal structure of the minimal RNA-binding region of hStau1, consisting of dsRBDs 3 and 4, in complex with the 19 bp SBS from *ARF1* mRNA. The structure, at 2.9 Å resolution, shows that the tandem dsRBDs of hStau1 recognize A-form RNA, mainly through electrostatic interactions with the RNA backbone. A comparison with the structure of unbound *ARF1* SBS, at 1.9 Å resolution, reveals that the target RNA undergoes minor changes upon hStau1 binding. In addition to the RNA backbone interactions, we observe residues from both dsRBDs making direct contact with G and C bases within the minor groove of the target RNA, suggesting some sequence specificity. These base-contacting residues are conserved in *Drosophila* Stau, and are required for protein function in vivo. Our data show that both dsRBDs 3 and 4 of Stau are important for RNA binding in vitro and in vivo, and provide insights into target requirements in Stau recognition.

# Results

## dsRBDs 3 and 4 in hStau1 are sufficient for RNA binding in vitro

To better characterize RNA binding by hStau1, we used fluorescence anisotropy (FA) and determined the binding affinity of the full-length (FL) protein, and truncated versions of the protein, for *ARF1* SBS. Purified recombinant hStau1 FL interacts with *ARF1* SBS with high affinity (6 ± 1 nM), but binds a single-stranded version of the same site (ss*ARF1*), used as a control for nonspecific protein–RNA interaction, with a 16-fold lower affinity (96 ± 27 nM) (Fig 1D). A C-terminal deletion, removing SSM and dsRBD5, or an N-terminal deletion, removing dsRBD2, only minimally affects the RNA-binding affinity (10 ± 2 and 9 ± 3 nM, respectively). A hStau1 fragment consisting of dsRBDs 3 and 4 binds to *ARF1* SBS with an affinity of 14 ± 4 nM, in a similar range as that of the FL protein, with only a moderate (twofold) reduction.

We then tested the RNA-binding affinities of dsRBD3 and dsRBD4 separately, using GST-tagged constructs to increase the signal-to-noise ratio of the anisotropy measurements. A GST-tagged

dsRBD3-4 tandem construct binds *ARF1* SBS with similar affinity as its untagged version (9 ± 2 vs 14 ± 4 nM), indicating that the presence of a GST tag does not affect RNA binding (Fig 1D and E). Both dsRBD3 and dsRBD4, in isolation, bind *ARF1* SBS with drastically reduced affinities (10- and 1,000-fold, respectively), pointing to cooperativity within the tandem construct (Fig 1E).

These experiments show that a hStau1 construct including dsRBDs 3 and 4 is sufficient to mediate high affinity binding to the target dsRNA and, consistent with previous qualitative observations (Wickham et al, 1999; Micklem et al, 2000; Luo et al, 2002), dsRBD3 has a stronger contribution to the binding affinity.

## Protein- and RNA-mediated dimerization of hStau1 in solution

We sought to determine the stoichiometry of the hStau1-*ARF1* SBS complex in solution. First, we analyzed the multimerization state of the apo protein, either FL or with various truncations. Consistent with previous reports (Martel et al, 2010; Gleghorn et al, 2013), the presence of dsRBD5 and SSM promotes dimerization, as shown by size-exclusion chromatography (SEC) and multi-angle laser light scattering (MALLS) experiments (Figs 2A and S2A and C). However, when we analyzed hStau1 stoichiometry in the presence of *ARF1* SBS, we observed dimerization for all protein truncations, regardless of the presence of the dimerization domain, as shown by SEC–MALLS and by RNA bands supershift upon increasing protein concentrations in electrophoretic mobility shift assay (EMSA) experiments (Figs 2B and C, S2A–D). Therefore, the interaction with dsRNA is sufficient to promote dimerization of the hStau1 minimal RNA-interacting region independently from protein-mediated self-association.

## Structure determination of hStau1 dsRBDs 3 and 4 in complex with *ARF1* SBS

The only structural information available on Stau interaction with the RNA has been provided by the NMR structure of dStau dsRBD3 bound to a stem-loop RNA (Ramos et al, 2000). The study characterized the RNA-binding interface on dStau, identifying protein residues required for RNA binding in vitro and for protein function in vivo. However, it used an artificially selected RNA sequence for in vitro assays and structural determination, and thus could not identify specific RNA features required for Stau recognition.

To better understand the molecular details of RNA recognition by Stau proteins, and investigate the role of the different RNA-binding dsRBDs in Stau, we determined the crystal structure of the complex between hStau1 dsRBD3-4 (aa 182–360) and *ARF1* SBS (Fig 3A–C). The structure, at 2.9 Å resolution, was solved by molecular replacement (MR), using as a search model an ensemble of superposed available structures of Stau dsRBD5 (PBD ID: 4DKK, 5CFF; Gleghorn et al, 2013; Jia et al, 2015) and an idealized dsRNA model. After few cycles of refinement, additional electron density was observed, which allowed manual positioning of a homology model of dsRBD4 (SWISS-MODEL, Waterhouse et al, 2018) in the electron density. The refined model has an $R_{free}$ of 24.0% and $R_{factor}$ of 21.7%, with good stereochemistry (Table 1). A difference Fourier map, calculated using phases from a derivative complex containing SeMet-substituted hStau1 dsRBD3-4, showed peaks at the expected Met

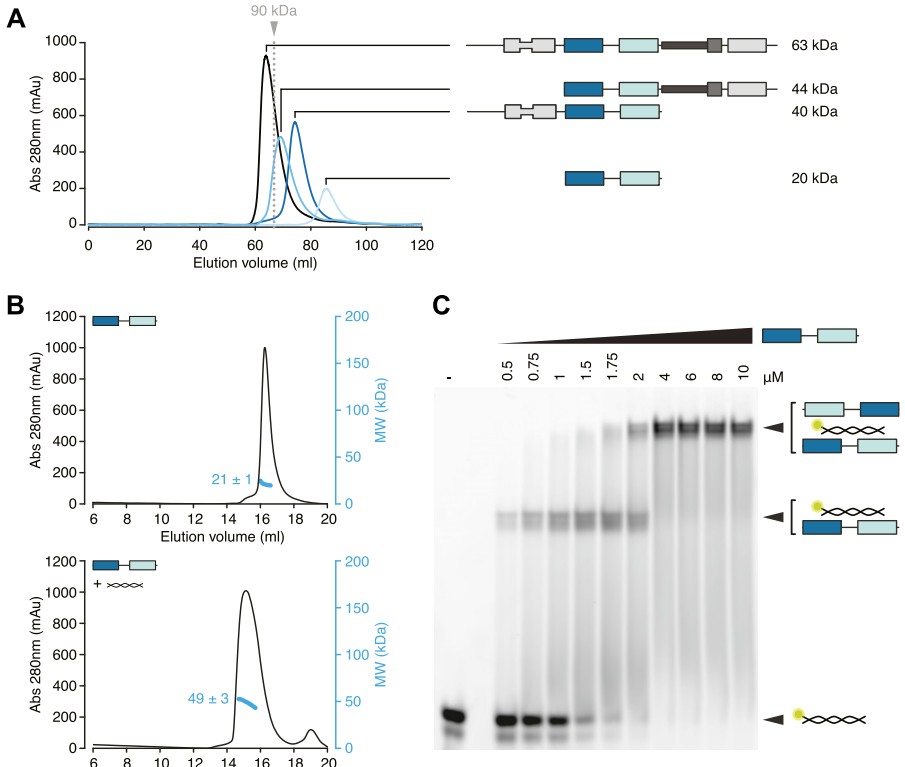

**Figure 2. dsRBDs 3 and 4 of hStau1 dimerize on the RNA.**
**(A)** Size exclusion chromatography (SEC) profile of the indicated hStau1 constructs. The calculated molecular weight (MW) for a monomer is depicted on the right. The elution volume of a reference globular protein is marked by a dotted gray line, with the corresponding MW on top. Protein constructs including SSM and dsRBD5 elute as dimers. **(B)** SEC profiles of purified hStau1 dsRBD3-4 alone (top), or pre-incubated with *ARF1* SBS (bottom). Calculated MW, obtained from MALLS, is indicated in blue. **(C)** EMSA of 5′-fluorescein-labeled *ARF1* SBS titrated with hStau1 dsRBD3-4 construct. Protein concentrations (in μM) are indicated above the gel; RNA concentration was 1 μM.

positions in all three domains (M181 and M213 for dsRBD3, and M321 and M348 for dsRBD4) (Fig S3A). This allowed the validation of the correct placement and assignment of the domains and confirmed the presence of two dsRBD3 domains (designated dsRBD3[A] and dsRBD3[B]), but only one dsRBD4 associated to one molecule of RNA. This dsRBD4 is not well defined and could explain the relatively

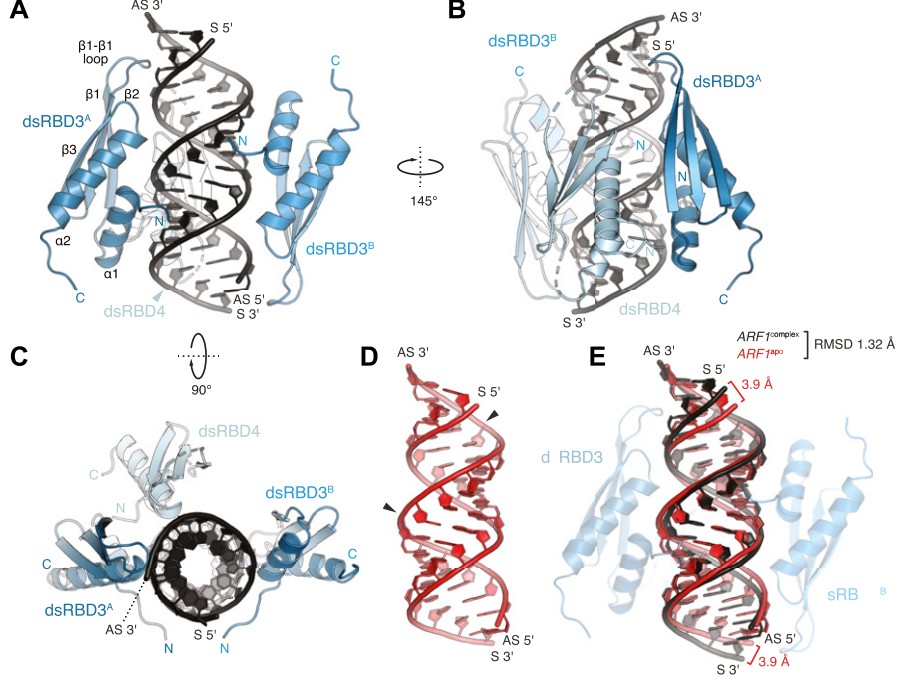

**Figure 3. Structure of hStau1 dsRBD3-4 in complex with *ARF1* SBS.**
**(A–C)** Structure of hStau1 dsRBD3-4 construct in complex with *ARF1* SBS, and in two views rotated by 145° about the vertical axis **(B)**, or 90° about the horizontal axis **(C)**. RNA (black), dsRBD3 (blue), and dsRBD4 (cyan) are shown as cartoons; dotted lines represent loop connections that are not visible in the electron density. **(D)** Structure of *ARF1* SBS in its unbound state, in the same orientation as in (A). G–U wobble base pairs are indicated by arrows. **(E)** Superposition of *ARF1* SBS in its unbound state (red) and in complex with hStau1 (black), with the corresponding root mean square deviation (RMSD) are indicated. All protein structure figures were generated using PyMOL v2.1 (http://www.pymol.org).

**Table 1.   Data collection and refinement statistics of the crystal structures of the hStau1-*ARF1* SBS complex and of *ARF1* SBS.**

| | Native *ARF1*-hStau1[182–360] | *ARF1*-hStau1[182–360] SeMet | *ARF1* SBS |
|---|---|---|---|
| Data collection | | | |
| Beamline | X06DA-PXIII | X06DA-PXIII | X10SA-PXII |
| Space group | $P4_12_12$ | $P4_12_12$ | $H32$ |
| Cell dimensions | | | |
| $a, b, c$ (Å) | 105.9, 105.9, 169.2 | 105.1, 105.1, 171.1 | 43.8, 43.8, 452.1 |
| $\alpha, \beta, \gamma$ (°) | 90, 90, 90 | 90, 90, 90 | 90, 90, 120 |
| Wavelength | 1.045 | 0.980 | 1.000 |
| Resolution (Å) | 50–2.89 (3.06-2.89) | 50–4.23 (4.48–4.23) | 38–1.9 (2.02–1.90) |
| $R_{sym}$ or $R_{merge}$ | 0.133 (2.864) | 0.274 (1.685) | 0.089 (1.341) |
| $I/\sigma I$ | 20.52 (1.32) | 10.81 (2.03) | 13.52 (1.70) |
| Completeness (%) | 99.4 (96.8) | 99.7 (99.0) | 99.9 (99.6) |
| Redundancy | 27.5 (25.5) | 27.7 (25.5.0) | 10.3 (10.3) |
| CC (1/2) | 100 (81.4) | 100 (75.1) | 100 (69.5) |
| Refinement | | | |
| Resolution (Å) | 47.35–2.89 | | 38–1.9 |
| No. reflections | 40,874 | | 13,828 |
| $R_{work}/R_{free}$ | 21.7/24.0 | | 24.4/26.2 |
| No. atoms | 4,278 | | 1,883 |
| Protein | 3,058 | | |
| RNA | 1,217 | | 1,828 |
| Water | 3 | | 40 |
| $B$-factors (Å$^2$) | 109.7 | | 54.4 |
| Protein | 126.7 | | |
| RNA | 77.4 | | 53.5 |
| Water | 73.2 | | 46.7 |
| Root mean square deviations | | | |
| Bond lengths (Å) | 0.004 | | 0.019 |
| Bond angles (°) | 0.58 | | 2.38 |

One native crystal for each construct and one SeMet crystal were used for data collection. Values in parentheses are for the highest-resolution shell.

high B-factors of this portion of the structure. One dsRBD4 is disordered and could not be modeled (Fig 3A–C). In the final model, apart from the disordered dsRBD4, five residues at the C-terminus and two loop regions within the visible dsRBD4 are disordered (Fig 3B and C). The linker region between the two dsRBDs, present in the crystallized complex (Fig S3B) is disordered and could not be modeled. The *ARF1* SBS RNA duplex could be fully modeled.

### Overview of the structure

Both hStau1 dsRBDs 3 and 4 adopt the classical $\alpha\beta\beta\beta\alpha$ topology, with two $\alpha$-helices packed to a three-stranded antiparallel $\beta$-sheet (Fig 3A); the structures are very similar to the ones of dStau dsRBD3 (Bycroft et al, 1995; Ramos et al, 2000) and murine Stau dsRBD4 (PDB ID: 1UHZ, unpublished) in isolation. For each domain, the interaction surface with the RNA spans the major groove and the two adjacent minor groove surfaces. DsRBD3[A] and dsRBD4 bind to the

RNA next to each other, sharing a heterodimerization interface of 273 Å$^2$ (5.1% of their total surface) (calculated with PISA; Krissinel & Henrick, 2007), which engages strand $\beta1$ of dsRBD3[A] and helix $\alpha2$ of dsRBD4 (Figs 3B and S3C). Mutations designed to interfere with this interface have no effect on the RNA-binding affinity, suggesting that this interaction is not required for binding cooperativity between the dsRBDs (Fig S3D). DsRBD3[B] is bound on the opposite side of the RNA molecule, in an antiparallel orientation to dsRBD3[A], whereas the second dsRBD4 is missing and likely to be unbound and disordered, possibly because of steric clashes with the other bound dsRBDs (Fig 3A and C).

A Stau-dsRNA complex with the 2:1 stoichiometry observed in the structure also forms in solution with the same components (Fig 2B and C). Stau dimerization is known to occur both in vitro and in vivo; nevertheless, the dsRBDs arrangement in our structure might be influenced by the use of a truncated protein construct that, although binding dsRNA with similar affinity as wt Stau, could leave a smaller fingerprint on the RNA than the full-length protein. The

length of the dsRNA substrate and crystal packing forces could also induce the binding mode observed.

Based on sequence length, the linker between dsRBDs 3 and 4 is predicted to span a range of 100–120 Å, a distance that is compatible with a connection of dsRBD4 with either dsRBD3[A] or dsRBD3[B]. Shortening of this linker, from 29 to 8 residues, has a strong effect on RNA binding (eightfold reduction; Fig S3D). However, mutating four of seven positively charged residues in this region also reduces RNA-binding affinity (threefold reduction; Fig S3D), suggesting that some residues in the linker are involved in stabilizing interactions with the RNA.

### hStau1 binding site on *ARF1* RNA is an A-form double helix

To test if binding of hStau1 induces conformational changes in the target RNA, we determined the structure of the *ARF1* SBS (Figs 1C and S4) in isolation, at 1.9 Å resolution. Phase information was obtained by MR, using the available model of an RNA duplex (PDB ID: 1QC0; Klosterman et al, 1999) as query. The crystals belong to space group *H*32, with the asymmetric unit (ASU) including one and half RNA molecules, co-axially stacked. This arrangement generates a sixfold static disorder, as previously observed in crystal structures of dsRNAs of similar length (Klosterman et al, 1999). The refined *ARF1* SBS model has an $R_{\text{free}}$ of 24.8% and $R_{\text{factor}}$ of 22.6%, with good stereochemistry (Table 1). The *ARF1* SBS forms a blunt end duplex with standard A-form helix conformation and Watson–Crick base pairing throughout, except for two wobble base pairs (Fig 3D).

### Minor conformational changes occur in the RNA upon hStau1 dsRBD3-4 binding

We do not observe major conformational changes in *ARF1* SBS upon hStau1 binding. The structural models of apo and bound *ARF1* SBS

superpose well, with 1.32 Å root mean square deviation (Fig 3E). However, we detect minor changes in several regions of the RNA molecule. Overall, bound *ARF1* SBS is slightly more stretched than in its unbound state, resulting in a slight widening of the major groove at points engaged in interaction with the helix α2 of dsRBD3[A] and dsRBD3[B] (Fig S5A and B). Moreover, small displacements of ~4 Å take place at both 3′ and 5′ end of the *ARF1* sense strand, in regions contacted by the β1–β2 loop of dsRBD3[A] and dsRBD3[B] (Figs 3E and S5A and B). In addition, a minor deviation from RNA co-linearity stacking occurs at the contact point of dsRBD3[B] with the symmetry-related RNA molecule (Fig 4A). This deviation mimics a major/minor groove arrangement and engages loop β1–β2 of dsRBD3[B]. It is unclear whether the observed minor deviations from A-form dsRNA in bound *ARF1* SBS are due to protein binding or the coaxial stacking arrangement of the duplex in the crystal lattice.

### Sugar-backbone interactions drive binding

Typically, dsRBDs interact with the dsRNA backbone using polar and basic residues clustered in three distinct regions (Masliah et al, 2013). Region 1, consisting of helix α1, contacts the minor groove of the dsRNA; region 2, formed by the loop between strands β1 and β2, interacts with the adjacent minor groove surface; region 3, comprising a patch of basic residues at the N-terminal tip of helix α2, inserts in the intervening major groove (Fig 3A). Both dsRBD3 molecules in our structure engage all three regions in RNA binding (Fig 4A). This binding mode is similar to what was observed for dStau dsRBD3 (Ramos et al, 2000) and other dsRBD–dsRNA complexes (Ryter & Schultz, 1998; Blaszczyk et al, 2004; Wu et al, 2004; Gan et al, 2006, 2008; Stefl et al, 2006, 2010; Wang et al, 2011; Jayachandran et al, 2016; Masliah et al, 2018). Conversely, dsRBD4

**Figure 4. Interactions between dsRBDs 3 and 4 of hStau1 and *ARF1* SBS.**
**(A)** Close-up view of hStau1 dsRBD3[A] (left), dsRBD3[B] (middle), and dsRBD4 (right) in complex with *ARF1* SBS. Residues making contact with the RNA are shown as sticks and labeled; residues making base contacts within the minor groove of the dsRNA are marked in bold; residues mutated in (B) are highlighted in red. **(B)** The scheme on the left depicts the three dsRBD regions interacting with dsRNA: helix α1 (region 1) and loop β1–β2 (region 2) bind two subsequent minor grooves, whereas the N-terminal half of helix α2 (region 3) contacts the intervening major groove. Right: $K_d$ values determined by FA, as in Fig 1D and E. Mutated residues, in region 2 and 3, are labeled in red in (A).

primarily uses regions 1 and 3 for binding, whereas the β1–β2 loop in region 2 is partially unstructured and does not engage in close interactions with the RNA (Fig 4A). Mutating conserved residues in two of the three binding regions in dsRBD3 (H212A and K214A in region 2; K234E, K235A, and K238A in region 3), while not affecting protein folding (Fig S6C), causes a strong (eightfold) reduction in the affinity for *ARF1* SBS (Fig 4B). Corresponding mutations in dsRBD4 (R315A and R317A in region 2; K337E, K338A, and K341A in region 3) have a 4.5-fold effect on RNA-binding affinity, further supporting a stronger contribution of dsRBD3 to substrate binding. These and all other protein mutants used for in vitro assays are folded (Fig S6A–E).

## Base interactions are involved in substrate recognition

In addition to the interactions with the sugar–phosphate backbone, both domains of hStau1 in our structure directly contact RNA bases in the minor groove of *ARF1* SBS (Fig 5A). These contacts involve helix α1 (region 1) and the β1–β2 loop (region 2).

Within the β1–β2 loop of dsRBD3[A], the backbone oxygen of P211 establishes a base-specific contact with the N2 group of G77 (Figs 5E and S7C); the same loop is reoriented in dsRBD3[B] and points toward the minor groove of the symmetry-related RNA molecule without engaging in base-directed contacts (Fig 4A), whereas in dsRBD4, the β1–β2 loop is not in close contact with the minor groove (Fig 4A). The

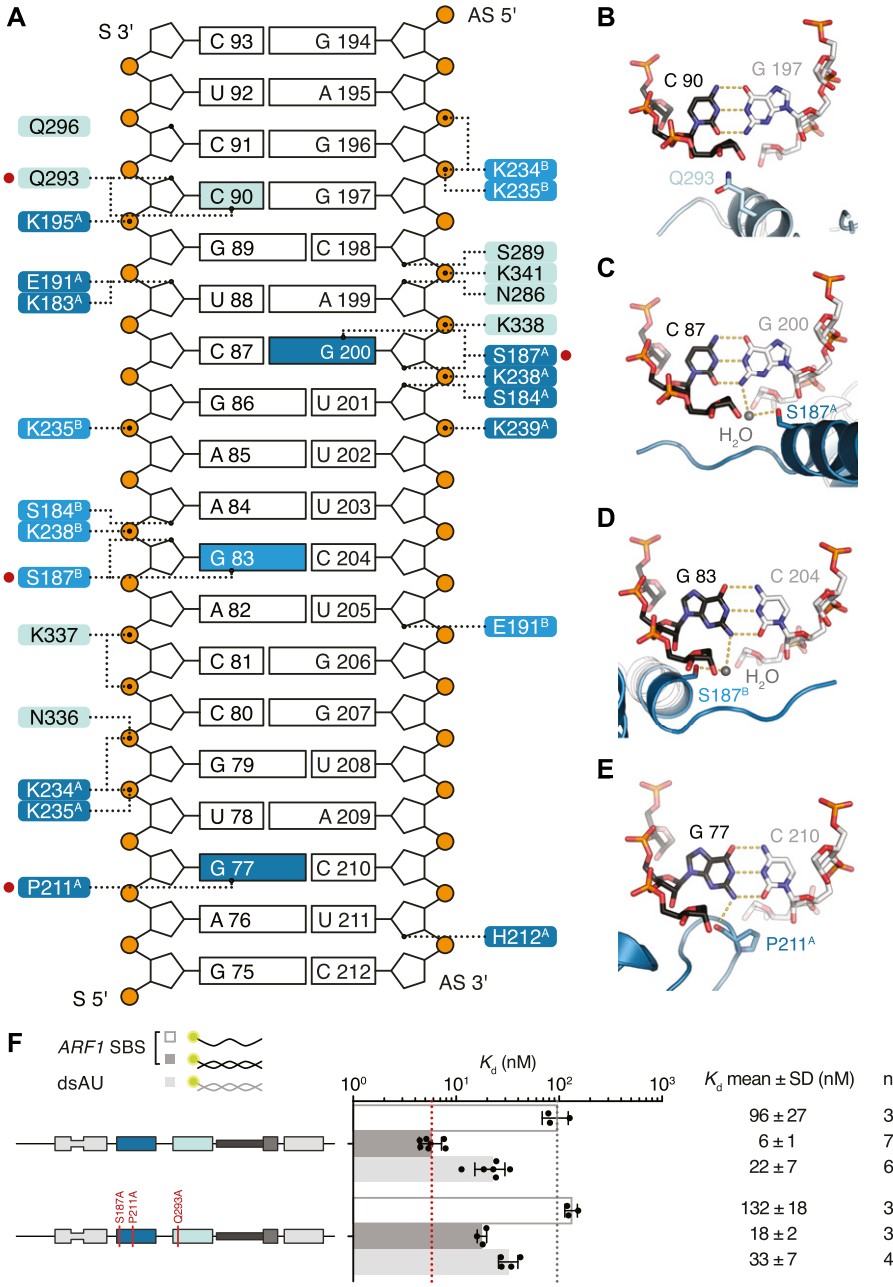

**Figure 5. Residues in dsRBDs 3 and 4 make base-directed contacts with *ARF1* SBS.**
**(A)** Schematic representation of *ARF1* SBS. Dotted lines indicate contacts between residues in hStau1 dsRBD3[A] (dark blue), dsRBD3[B] (blue), or dsRBD4 (cyan), and the RNA. Red dots mark residues interacting with the RNA bases (colored in the scheme). **(B–E)** Detailed views of the base-directed interactions, with hydrogen bonds indicated as dotted lines. The view is down the RNA helix axis. Nucleotides of the *ARF1* SBS sense and antisense strand are in black and gray, respectively. **(F)** $K_d$ values determined by FA using 5′-fluorescein-labeled *ARF1* SBS, either single or double stranded, or a dsRNA of the same length and of a random AU sequence (dsAU).

| $K_d$ mean ± SD (nM) | n |
|---|---|
| 96 ± 27 | 3 |
| 6 ± 1 | 7 |
| 22 ± 7 | 6 |
| 132 ± 18 | 3 |
| 18 ± 2 | 3 |
| 33 ± 7 | 4 |

contact established by P211 is "structural", as it involves the backbone oxygen and is observed in most dsRBDs (Ryter & Schultz, 1998; Blaszczyk et al, 2004; Gan et al, 2006, 2008; Stefl et al, 2010; Yang et al, 2010; Masliah et al, 2013; Jayachandran et al, 2016; de Morrée et al, 2017) (Fig S8). The interaction selects specifically a G base but, given the central position of the N2 group in the minor groove, it could be formed within either a G–C or C–G base pair (Seeman et al, 1976; Steitz, 1990). This isolated contact might not be discriminatory, but it could serve as a hook to bind to the first minor groove.

In helix α1, S187 makes a similar water-mediated contact in both dsRBD3 molecules present in the complex, interacting with the N2 group of G200 (dsRBD3[A]) or G83 (dsRBD3[B]) (Figs 5C and D and S7A). It is possible that a different nucleotide could be contacted through repositioning of the water molecule; nevertheless, there are examples of water-mediated interactions between protein and nucleic acids that have been proven critical for specific target recognition (Seeman et al, 1976; Otwinowski et al, 1988; Rould et al, 1989, 1991; Shakked et al, 1994; Wilson et al, 1995; Kosztin et al, 1997).

The electron density for dsRBD4 is less defined than that of dsRBD3; however, the side chain of Q293, within helix α1, is directed toward the edge of C90 (4.4 Å distance), suggesting a base-directed interaction (Figs 5B and S7B).

All base-directed contacts observed in the complex engage G–C base pairs; although none of the contacts are unequivocal alone, the combination of the three and the spacing between them are likely to impose constraints on the binding mode and guide the protein to the correct position on the dsRNA. To test the importance of these interactions in hStau1 target recognition, we used FA to measure binding affinity of hStau1 to a dsRNA of the same length as ARF1 SBS, but of a random AU sequence (dsAU). We decided to use a dsRNA without any G–C pairs to prevent alternative binding

orientations that would be possible on ARF1 SBS (Fig 6A). We observed about a fourfold reduction in binding affinity, suggesting that interactions with G–C base pairs increase hStau1 RNA-binding affinity (Fig 6B). A hStau1 protein carrying mutations designed to affect base-directed contacts (S187A, P211A, and Q293A) binds dsAU with a similar affinity (1.5-fold reduction), but shows a stronger (threefold) reduction in binding ARF1 SBS when compared with the wt protein (Fig 5F). We also tested hStau1 binding to dsAU containing one, two, or three G–C base pairs at positions where we observed base-directed contacts to ARF1 SBS (Fig 6A and B). The addition of the G–C pairs slightly increases hStau1 binding affinity. A dsRNA containing three G–C pairs, at positions compatible with the interaction of dsRBD3[A] (with a register of G–X$_9$–G) and dsRBD4, binds with a similar affinity as ARF1 SBS, supporting the role of G–C pairs in target recognition (Fig 6A and B). Our results indicate that these contacts influence RNA binding in a sequence-dependent manner.

## Base interactions affect phenotypic outcome

Our structural and biochemical results show that hStau1 target recognition depends not only on RNA shape, but also on sequence. Residues involved in RNA-binding, including those engaging in base-directed contacts, are conserved in Stau family members, including *Drosophila* Stau. Furthermore, known binding sites of dStau, such as *bcd* III and Vb, as well as previously characterized artificial RNA duplexes (Fig S9), contain G–C pairs in a suitable register. To test the conservation and functional relevance of shape and sequence-mediated RNA recognition in vivo, we generated *Drosophila* strains expressing GFP-tagged dStau, wt, or mutants in a Stau protein–null background. All transgenes are expressed in the

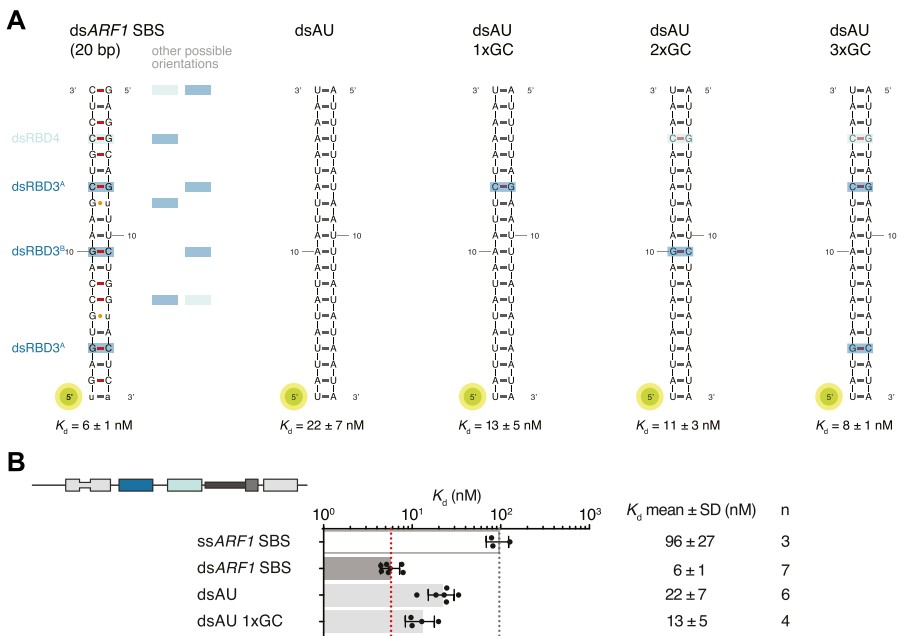

**Figure 6. G–C addition to a synthetic dsRNA substrate restores affinity of binding.**
**(A)** Schematic representation of the ARF1 SBS and of the designed synthetic duplexes. Watson–Crick base pairs are labeled with an hyphen in different colors with C–G in red and U–A in gray, whereas wobble pairs with a yellow dot. Below each RNA, the $K_d$ of binding of hStau1 FL is indicated. Boxes highlight base contacts by dsRBD3 (in blue) and dsRBD4 (in cyan). **(B)** $K_d$ values determined by fluorescence anisotropy (FA), using 5′-fluorescein-labeled dsRNAs. The recombinantly purified hStau1 FL is indicated in the schematic above. The graphs show mean $K_d$ (bars), standard deviation (black lines), and $K_d$ values obtained in each independent experiment (black dots). Mean $K_d$ ± SD, in nM, and number of independent measurements (n) are indicated on the right.

maternal germline under the control of the α4-tubulin promoter, inserted in the same locus and equally expressed in ovaries (Fig 7A and B). As previously reported, embryos generated by Stau-null mothers have severe anterior–posterior patterning defects, because of the mislocalization of *bcd* and *osk* mRNAs, which can be rescued by the expression of GFP-dStau (St Johnston et al, 1991; Micklem et al, 1997, 2000) (Fig 7). Mutation of RNA backbone-contacting residues in dsRBD3 (H606A and K608A in region 2; K628E and K629A in region 3) fails to rescue the Stau-null phenotype in vivo (Ramos et al, 2000) (Fig 7) and strongly impairs binding to *bcd* stem-loop III in vitro (almost 10-fold reduction) (Fig 7C). Corresponding mutations in dsRBD4 (R742A and R744A in region 2; K764E, K765A, and K768A in region 3) have a milder effect on RNA-binding affinity (less than threefold reduction) and retain the ability to rescue the Stau-null phenotype. We also generated point mutations in dStau targeting residues involved in direct base contacts in hStau1 (Figs 5F and 7C). A single mutation in helix α1 of dsRBD4 (Q718A, corresponding to Q293A in hStau1) shows a similarly weak effect in RNA-binding affinity in vitro as that of a mutant that perturbs a backbone-mediated interaction. However, this mutant rescues less efficiently than the backbone-mediated interaction mutant in vivo, suggesting that this residue plays a role in substrate recognition (Fig 7A). The additional mutation of a residue involved in direct sequence interaction in helix α1 of dsRBD3 (S581A, corresponding to S187A in hStau1) has a moderate effect on *bcd* III binding in vitro (3.5-fold reduction of affinity when compared with the wt), but dramatically impairs protein function in vivo. These results show that the binding affinity measured in vitro is not strictly predictive of Stau function in vivo, and show that base-directed interactions might be required for in target recognition in vivo.

# Discussion

Here, we combined crystallography and structure-based mutagenesis to dissect the determinants of RNA recognition by Stau, and tested the in vivo relevance of identified key Stau residues in *Drosophila* rescue experiments.

In the complex, Stau dsRBDs showed a canonical binding mode to the RNA backbone, with some variations. The association of two dsRBD3 molecules to the same substrate and their differential engagement of the canonical contacts in the β1–β2 loop supports a versatility of binding that might be needed to allow Stau recognition of several RNA targets, possibly with different shapes. Indeed, a variety of Stau targets have been identified in vivo (St Johnston et al, 1991; Ferrandon et al, 1994; Li et al, 1997; Mallardo et al, 2003; Kim et al, 2005, 2007; Furic et al, 2008; Heraud-Farlow et al, 2013; Laver et al, 2013; LeGendre et al, 2013; de Lucas et al, 2014; Ricci et al, 2014; Sugimoto et al, 2015). dsRBD3 and dsRBD4 show different modes of binding, with dsRBD4 engaging only two of the three canonical regions in its interaction with the RNA. This suggests a model in which dsRBD3 would first associate with the target, facilitating the subsequent interaction of dsRBD4. Alternatively, Stau could use the weaker binder, dsRBD4, to scan the pool of RNAs and lock on targets through the succeeding binding of dsRBD3. This step-wise manner of recognition will need to be further explored through studies of binding dynamics.

A conformational flexibility of the β1–β2 loop has also been observed by NMR in the dsRBD3 of dStau (Ramos et al, 2000; Castrignanò et al, 2002) and in the dsRBDs from other proteins, such as Xlrbpa (Ryter & Schultz, 1998) (Fig S8). Another difference between the two dsRBDs is that dsRBD4 can interact with both ss and

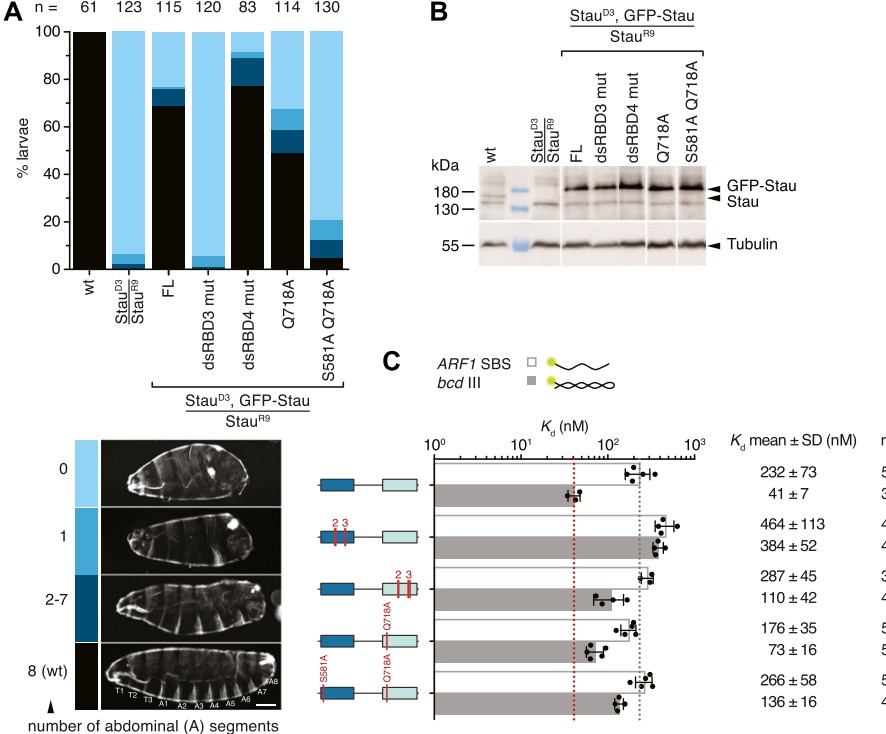

**Figure 7. Residues involved in base-directed contacts are important in vivo in *Drosophila*.**
**(A)** Analysis of cuticle preparations of 24-h-old *Drosophila* larvae; the genotype of the mother is indicated below the graph and the number of counted larvae on top. The color coding legend is shown below; larvae were classified according to the number of abdominal segments. In the images, anterior is to the left and dorsal on top. T1–T3, thoracic segments; A1–A8, abdominal segments. Scale bar: 100 μm. **(B)** Western blot showing the expression levels of endogenous dStau, of *stau*[R9]/*stau*[D3] (protein null), and of the transgenic GFP-dStau constructs, in ovaries of flies with the indicated genotype. The blot was probed with anti-Stau serum (top panel); anti-α-tubulin (bottom panel) was used as a loading control. **(C)** $K_d$ values determined by FA using 5′-fluorescein-labeled *ARF1* SBS (sense strand), or stem-loop III from *bcd* 3′UTR, and the indicated dStau protein constructs. Residues mutated in dStau correspond to residues involved in RNA interaction (either nonspecific or base-directed) in hStau1 (Figs 4 and 5, and S1B); P211 in hStau1 corresponds to A605 in dStau.
Source data are available online for this figure.

dsRNA (Fig 1E). In vivo, dsRBD4 may thus bind different RNA secondary structures containing both ss and dsRNA stretches, such as loops and bulges. A similar domain specialization has been described in the adenosine deaminase ADAR2 where one of the dsRBDs binds hairpin loops, whereas the other prefers duplexes (Stefl et al, 2006, 2010).

We showed that the dsRBDs of Stau have different contributions to binding affinity and specificity. In hStau1, both dsRBD 3 and 4 can bind dsRNA independently in vitro; however, efficient target binding requires both domains. A similar cooperativity was also observed for other proteins containing multiple RNA-binding domains (Shamoo et al, 1995; Tian & Mathews, 2001; Chang & Ramos, 2005; Lunde et al, 2007; Parker et al, 2008). In other Stau proteins, such as hStau2 and dStau, dsRBD1 could add further affinity and specificity to target recognition.

Consistently with its stronger contribution to RNA binding, reducing dsRBD3 affinity for the RNA by mutating residues that interact with the RNA backbone has a dramatic effect on dStau activity in vivo. In contrast, mutations in backbone-interacting residues within dsRBD4 do not significantly affect the phenotypic outcome. One caveat of these rescue experiments is that the proteins are slightly overexpressed relative to the endogenous levels, which may compensate for a certain reduction in RNA-binding affinity, provided that specificity is maintained. We can nevertheless compare the different mutant forms in vivo, because we used the same genomic integration site for all transgenes.

There has been a discrepancy between in vivo data showing that Stau binds specifically to RNA targets, with striking examples of paradigmatic localized mRNAs in *Drosophila* such as *osk*, *bcd*, and *prospero* (Broadus and Doe, 1997; Broadus et al, 1998; Ephrussi et al, 1991; Ferrandon et al, 1994; Fuerstenberg et al, 1998; Kim-Ha et al, 1991, 1995; Li et al, 1997; Matsuzaki et al, 1998; Schuldt et al, 1998; Shen et al, 1998; St Johnston et al, 1989, 1991, 1992), and in vitro data that show the limited specificity of binding (St Johnston et al, 1992; Marión et al, 1999; Wickham et al, 1999; Ramos et al, 2000; LeGendre et al, 2013; Wang et al, 2015). Our structural analysis suggests few direct contacts of Stau side chains with RNA bases. A contact at loop $\beta1$–$\beta2$, a feature shared by most dsRBDs (Masliah et al, 2013), and two contacts in helix $\alpha1$ suggest preference for substrates with G–C base pairs in two adjacent minor grooves. When the residues involved in base interactions are mutated, there is a minimal effect on RNA-binding affinity, as judged by in vitro measurements. An effect of similar magnitude on RNA binding is observed when adding G–C base pairs at the observed register length distance to a pure A–U RNA duplex. Base-directed contacts at equivalent structural positions within helix $\alpha1$ have been observed in other dsRBDs, but involving different residues and potentially giving rise to a different sequence preference (Ryter & Schultz, 1998; Blaszczyk et al, 2004; Gan et al, 2008; Stefl et al, 2010; Jayachandran et al, 2016) (Fig S8). Mutations designed to impair base-directed interactions in either dsRBD4 or in dsRBDs 3 and 4 impair in vivo dStau function with a strong additive effect, indicating that these residues in both domains are required and may contribute to target recognition. The same mutations only mildly affect RNA-binding affinity in vitro, challenging the assumption that in vitro affinity measurements can predict in vivo activity. Thus, our results highlight the importance of combining multiple approaches to study the function of RNA-binding proteins.

We show that base interactions are relevant in vivo and we think that they contribute to the overall sequence selectivity by Stau, possibly together with additional regions of the protein or with other regulators. In particular, the linker connecting dsRBDs 3 and 4 is flexible and could allow different positioning of the dsRBDs relative to each other, enabling the recognition of targets of different sequence and structure. The linker could also permit *trans* interactions, bridging multiple RNAs, or *cis* binding to either adjacent or non-adjacent sites within a target, such as loops III, IV, and V in *bcd* 3′UTR.

Our results suggest a model in which Stau can strongly bind to dsRNA through non-sequence specific interactions. Correct positioning of the Stau RNA-binding module might follow the scanning of potential specific binding motifs, for example by sliding along dsRNA, as observed for hStau1 (Wang et al, 2015). Upon reaching a specific binding site, containing correctly spaced G–C base pairs in the minor groove or perhaps complex secondary structure elements with both ss and dsRNA sites, such as the "apex" site adjacent to the *ARF1* SBS (Fig 1C) (Kim et al, 2007), the different and flexibly linked dsRBDs of Stau may engage in base-pair interactions critical for in vivo function.

A dsRBD-containing protein, adenosine deaminase, acting on RNA 1 (ADAR1), inhibits SMD by competing with hStau1 binding to dsRNAs in 3′UTRs (Sakurai et al, 2017). From sequence analysis, helix $\alpha1$ of dsRBD1 of ADAR1 presents similar base-contacting residues as Stau dsRBDs 3 and 4 (Fig S1B). It would be interesting to explore if ADAR1 would have similar preference as Stau. Moreover, identification and better characterization of the RNA targets in *Drosophila* early development, together with a precise mapping of the binding sites on the RNAs, would be needed for a comprehensive understanding of the mechanisms underlaying our observations in vivo.

## Materials and Methods

### Protein expression and purification

The sequence of hStau1 (GenBank: AAD17531.1), codon optimized (Genscript), and of dStau (GenBank: AAA73062.1) were cloned in a pET-MCN vector (Fribourg et al, 2001). Stau constructs were cloned either in-frame with an N-terminal GST tag or with a hexahistidine (His) tag. Mutants and truncations were generated by site-directed mutagenesis. The recombinant proteins were expressed in *E. coli* BL21 (DE3) Star (Life Technologies) cells in Terrific broth medium, overnight at 20°C. For SeMet-substituted protein expression, *E. coli* DL41 (DE3) cells were used; cells were grown in M9 medium and starved before SeMet addition.

His-tagged Stau constructs were purified by cobalt affinity chromatography in lysis buffer (20 mM Tris–HCl, pH 7.5, at 4°C, 1.2 M NaCl) containing 0.5 mM DTT. After washing in buffer A (20 mM Tris–HCl, pH 7.5, at 4°C, 300 mM NaCl), the recombinant protein was eluted from the resin with a gradient to 300 mM imidazole. The His-tag was subsequently removed by dialysis in the presence of tobacco etch virus (TEV) protease overnight at 4°C, in dialysis buffer (20 mM Tris–HCl, pH 7.5, at 4°C, 100 mM NaCl, 1 mM DTT). GST-tagged

**Life Science Alliance**

hStau1 constructs were affinity-purified on glutathione resin in lysis buffer containing 1 mM DTT. After washing in buffer A supplemented with 1 mM DTT, the recombinant protein was either eluted from the resin with a gradient to buffer B (buffer A plus 40 mM reduced glutathione) or the tag was cleaved by TEV protease in batch overnight at 4°C. If required, proteins were further purified on HiTrap CaptoS (hStau1) or Heparin Sepharose 6 Fast Flow (dStau) columns (GE Healthcare), eluted with a gradient to 1 M NaCl, and finally applied on a HiLoad 16/600 Superdex 200pg column (GE Healthcare). For RNA-binding assays, proteins were gel filtrated in buffer H (20 mM Hepes, pH 7.5, at RT and 150 mM KCl). For crystallization, the complex was reconstituted with a 1.2 M excess of *ARF1* SBS dsRNA (Integrated DNA Technologies) added to recombinant hStau1 at RT, incubated for 1 h at 4°C, and loaded on a size exclusion column (20 mM Tris, pH 7.5, at 4C, 100 mM KCl, and 1 mM DTT).

Construct boundaries for hStau1 truncations: aa 1–360, aa 182–577, aa 182–360; aa 182–255 (dsRBD3 alone); aa 284–358 (dsRBD4 alone). Boundaries for dStau dsRBD3-4 construct: aa 570–786.

### Crystallization, data collection, and analysis

Crystals of hStau1[182–360] (dsRBD3-4 construct) in complex with *ARF1* SBS grew at 22°C by vapor diffusion (50 mM MgCl₂, 120 mM KCl, 50 mM sodium cacodylate, pH 6.33, 5% 1,6-hexanediol). For data collection, crystals were cryo-protected with mother liquor supplemented with 30% 1,6-Hexanediol and flash frozen in liquid nitrogen. The crystals diffracted to 2.9 Å resolution, belong to the space group $P4_12_12$ with cell dimensions of a = b = 105.9 Å, c = 169.2 Å, $α = β = γ = 90°$ and contain one complex in the ASU. Data were indexed, integrated, and scaled using XDS (Kabsch, 2010). The structure was solved by MR using PHASER (McCoy et al, 2007). Iterative cycles of model building and restrained refinement were carried out in COOT and PHENIX (Adams et al, 2010; Emsley et al, 2010). A single-wavelength anomalous dispersion/diffraction (SAD) dataset of the *ARF1*-hStau1[182–360] complex with SeMet-substituted hStau1[182–360] was used to calculate an anomalous differences Fourier map using log-likelihood gradient maps in PHASER (McCoy et al, 2007) to find anomalous scatterers and verify the model.

Crystals of unbound *ARF1* SBS dsRNA were grown at 22°C by vapor diffusion (1.7 M (NH₄)₂SO₄ and 100 mM Tri-sodium citrate, pH 6.2). Cryo-protection was achieved by shortly transferring the crystals to mother liquor supplemented with 20% glycerol before freezing. The crystals diffracted to 1.9 Å resolution, belong to the space group H32 with cell dimensions of a = b = 43.8 Å, c = 452.1 Å, $α = β = 90°$, $γ = 120°$ and contain one and a half molecules in the ASU.

Diffraction data were collected at the PXIII (*ARF1*-hStau1[182–360] complex native and derivative) and PXII (*ARF1* SBS unbound) beamlines of the Swiss Light Source. 100% of the residues in each structure fall within the allowed regions of the Ramachandran plot.

### EMSA

For EMSAs experiments, *ARF1* SBS dsRNA was prepared by mixing equimolar amounts of 5'-6-fluorescein amidite (FAM)-labeled sense strand and unlabeled antisense strand (Integrated DNA Technologies). The annealed dsRNA, at a concentration of 1 µM, was incubated with increasing concentrations of purified hStau1 constructs in a final volume of 10 µl, in buffer H (see the Protein expression and purification section). After incubating for 45 min on ice, the samples were loaded on a 6% polyacrylamide gel and run at 220 V for 3–4 h in 1× TG buffer (25 mM Tris, 192 mM glycine, pH 9.78, at RT) at 4°C. The gel was then imaged at 488 nm using an Amersham Typhoon gel imager (GE Healthcare).

### MALLS

For MALLS, purified hStau1 and truncations were loaded onto a Superdex 200 Increase 10/300 GL column (GE Healthcare), and connected to a miniDAWN TREOS MALLS detector and Optilab T-rEX differential refractometer (Wyatt Technologies). Runs were performed in buffer (20 mM Tris–HCl, pH 7.5, and 150 mM KCl, at RT). For the proteins alone, 200 µg (FL), 400 µg (hStau1[1–360]), 430 µg (hStau1[182–360]), or 350 µg (hStau1[182–577]) protein were injected on the column. For RNA–protein complexes, 140 µg of wt or truncated protein were pre-incubated with a 1.2 M excess of *ARF1* SBS dsRNA. Molecular weight calculations were performed using ASTRA software (Wyatt Technologies).

### FA

FA measurements were performed with 5'-6-FAM-labeled RNA at RT (19–21°C) on an Infinite F200 plate reader (Tecan). The RNA, at a concentration of 10 nM, was incubated with different concentrations of purified hStau1 or mutants, in buffer H. The final reaction volume was 50 µl. Each titration point was measured three times per experiment, with an integration time of 40 µs, using 485 nm and 535 nm as excitation and emission wavelength, respectively. The data were analyzed using Prism 6 software (GraphPad) by nonlinear regression fitting to the following equation: $Y = Y_f + (Y_b - Y_f) \times (((L + K_d + X) - \sqrt{((L + K_d + X)^2) - (4 \times L \times X)))/(2 \times L))}$, where L = concentration of labeled RNA, X = protein concentration, $Y_f$ = anisotropy of free RNA = (X = 0), and $Y_b$ = anisotropy of bound RNA (Fluorescence Polarization Technical Resource Guide, fouth edition, Invitrogen).

The sequences of the RNA oligonucleotides (Integrated DNA Technologies) are as follows (5' to 3'): *ARF1* SBS sense (UGAGUGCCAGAAGCUGCCUC); *ARF1* SBS antisense (GAGGCAGUUUCUGGUACUCA); dsAU sense (UAUUAUAUAAAUUAUAAAAU); dsAU antisense (AUUUUAUAAUUUAUAUAAUA); and *bcd* III (CGCUAUUCGCCUUAGAUGUAUCUGGGUGGCUGCUCCACUAAAGCCCGGGAAUAUGCAACCAGUUACAUUUGAGGCCAUUUGGGCUUAAGCG).

### Circular dichroism (CD)

CD spectra were recorded on a Jasco J-810 spectropolarimeter, at 20°C. Measurements were performed using 200 µl of recombinantly purified hStau1 (wt or mutants), at a concentration of 0.2 mg/ml, in buffer C (20 mM Tris–HCl, pH 7.5, at 4°C, 150 mM KCl). Each spectrum was recorded five times and averaged.

### Fly stocks

The following fly stocks were used: Oregon-R (wt), $stau^{D3}$/SM6a, and $stau^{R9}$/SM6a. For the transgenes, a cDNA fragment of the *dStau*

gene (GenBank: AAA73062.1) was cloned into the pUmat-GFP-attB vector (GenBank: MH367504) via HindIII and NotI, yielding N-terminal GFP fusions of wt or mutated dStau proteins. The purified plasmids were injected into embryos from a recombinant stock with the genotype: $y^1$ M{vas-int.Dm}ZH-2A w*; PBac{y$^+$-attP-3B}VK00037 stau$^{D3}$/SM6a. Transgenic flies were identified in the F1 generation by the presence of orange eyes. Ovaries and embryos used for phenotype characterization were obtained from these transgenic flies crossed to stau$^{R9}$/SM6a flies, and selected against the presence of the balancer chromosome.

## Cuticle preparations

Early embryos were collected according to standard protocols (Schüpbach & Wieschaus, 1989), on apple-juice agar plates for 1 h, and allowed to age for 22–24 h at RT. Larvae were mounted in a 1:1 mixture of Hoyer's medium (A901A; Entomopraxis) and lactic acid, and cleared overnight at 70°C as described previously (Stern & Sucena, 2011). Images were acquired on a Zeiss AxioImager Z.1 microscope, in darkfield, using a Plan-Apochromat 10× objective (numerical aperture (NA) 0.45).

## Western blot and antibody staining

Ovaries were collected according to standard protocols (Palacios & St Johnston, 2002) and lysed in Laemmli Sample Buffer. The following antibodies were used for Western blots: anti-GFP polyclonal antibody (A11122, 1:2,000; Life Technologies), anti-$\alpha$-tubulin monoclonal antibody (T6074, 1:5,000; Sigma-Aldrich), and anti-Stau rabbit serum (St Johnston et al, 1991) (1:1,000).

## Accession codes

The coordinates and structure factors have been deposited in the Macromolecular Structure Database of European Bioinformatic Institute with ID code 6HTU and 6HU6 for hStau1-ARF1 SBS complex and ARF1 SBS, respectively.

# Supplementary Information

# Acknowledgements

We wish to thank the MPI-Tübingen and MPI-Martinsried Crystallization Facility. We are grateful to the staff at the Swiss Light Source synchrotron and Z Hong for assistance during data collection. We thank G Jékely and the Bono laboratory members for discussion and critical reading of the manuscript. This project received funding from the European Research Council (ERC) under the European Union's Seventh Framework Programme (FP7/2007-2013), ERC grant agreement no. 310957, and the Deutsche Forschungsgemeinschaft (FOR2333, BO3588/2-1 to F Bono).

## Author Contributions

D Lazzaretti: conceptualization, formal analysis, investigation, visualization, and writing—original draft, review, and editing.

L Bandholz-Cajamarca: investigation, visualization, methodology, and writing—original draft.
C Emmerich: investigation, methodology, and writing—original draft.
K Schaaf: investigation and methodology.
C Basquin: methodology.
U Irion: investigation and methodology.
F Bono: conceptualization, data curation, formal analysis, supervision, funding acquisition, validation, investigation, methodology, project administration, and writing—original draft, review, and editing.

## Conflict of Interest Statement

The authors declare that they have no conflict of interest.

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
