## [Reviewer comments · Life Science Alliance]

Crystal structure of Staufen1 with a physiological RNA sheds light on substrate selectivity

Daniela Lazzaretti, Lina Bandholz-Cajamarca, Christiane Emmerich, Kristina Schaaf, Claire Basquin, Uwe Irion, and Fulvia Bono

Corresponding author(s): Fulvia Bono, University of Exeter
DOI: 10.26508/lsa.201800187

Review timeline:

Submission Date:	2018-08-31
Editorial Decision:	2018-08-31
Revision Received:	2018-09-13
Editorial Decision:	2018-09-27
Revision Received:	2018-10-03
Accepted:	2018-10-05

Report:

(Note: Letters and reports are not edited. The original formatting of letters and referee reports may not be reflected in this compilation.)

Please note that the manuscript was previously reviewed at another journal and the reports were taken into account in the decision-making process at Life Science Alliance. Since the original reviews are not subject to Life Science Alliance's transparent review process policy, the reports and author response cannot be published.

1st Editorial Decision

31 August 2018

August 31, 2018

Re: Life Science Alliance manuscript #LSA-2018-00187-T

Fulvia Bono

Dear Dr. Bono,

Thank you for transferring your manuscript entitled "Crystal structure of Staufen1 in complex with a physiological RNA sheds light on substrate selectivity" to Life Science Alliance. The manuscript was assessed by expert reviewers at another journal before, and the editors transferred those reports to us with your permission.

The reviewers at the other journal appreciated that you provide the first crystal structure of a tandem dsRBD construct from Staufen1 with a natural substrate. However, they also thought that the biological relevance of your findings remains unclear at this stage, and that the conclusion that RNA binding by Staufen1 requires a sequence-specific RNA is not sufficiently supported. Both reviewers provided constructive input on how to revise your work to strengthen your conclusions. Based on these reports already at hand, we would like to offer publication of a revised version of your manuscript in Life Science Alliance. Please provide a point-by-point response to the concerns raised and accordingly text changes as well as more definitive proof for base-specific interactions (following the constructive input of both reviewers). We think that testing RNA mutant versions that probe specifically the highlighted GC base pair in the 3'-UTR, as suggested by Reviewer 2, is a

straightforward experiment that would elevate your paper significantly (or allow changing your interpretation).

Thank you for this interesting contribution to Life Science Alliance. We are looking forward to receiving your revised manuscript.

Sincerely,

2nd Editorial Decision

27 September 2018

September 27, 2018

RE: Life Science Alliance Manuscript #LSA-2018-00187-TR

Dr. Fulvia Bono
University of Exeter
Living Systems Institute
Stocker Road
Exeter EX4 4QD
United Kingdom

Dear Dr. Bono,

Thank you for submitting your revised manuscript entitled "Crystal structure of Staufen1 with a physiological RNA sheds light on substrate selectivity". The reviewers who evaluated your work at another journal before re-assessed this version, and as you can see below both reviewers appreciate the introduced changes and support publication in Life Science Alliance. We would thus be happy to publish your paper in Life Science Alliance pending final revisions necessary to address the comments made by reviewer # 2 and to meet our formatting guidelines.

- please update the manuscript to address the few comments made by reviewer #2
- please move the figure legends for supplementary figures into the docx file and upload the S figures as individual files not larger than a single page (=> please make sure that figure S6 is on a single page)
- please make sure that all figure panels are called out in the manuscript text (not all S figure panels are currently called out, Fig 5A and Fig 6B are not called out either)

A. FINAL FILES:

-- High-resolution figure, supplementary figure and video files uploaded as individual files: See our detailed guidelines for preparing your production-ready images, <http://life-science-alliance.org/authorguide>

B. MANUSCRIPT ORGANIZATION AND FORMATTING:

Full guidelines are available on our Instructions for Authors page, <http://life-science-alliance.org/authorguide>

Sincerely,

Andrea Leibfried, PhD
Executive Editor

Life Science Alliance
Meyershofstr. 1
69117 Heidelberg, Germany
t +49 6221 8891 502
e a.leibfried@life-science-alliance.org
www.life-science-alliance.org

Reviewer #1 (Comments to the Authors (Required)):

I am happy with the response to the reviewers and the additional data provided and the text changes. I do not have further request.

Reviewer #2 (Comments to the Authors (Required)):

The revision is improved. A few remaining issues are below.

1. p9, line 2 from the bottom, what is "(5,1% of the surface)"? and which surface?
2. p10, line 222, which figure "(Fig, B-C)"?
3. Many single-sentence paragraphs - p10, p13, p14, and p16 - should be avoided.
4. p13, line 297, make sure to include the "water-mediated" in the sentence: S187 makes a similar water-mediated contact in ...
5. p13, line 307, the side chain of Q293 is directed towards the edge of C90 - please include the information about the inter-atomic distance from the current structure, and include the sentences from Response to the Reviewers' Comments in the same paragraph of the main text: the side chain of Gln293 is not in hydrogen bonding with the base pair of C90-G197, and it does not establish a clear base-specific interaction but is in the vicinity of C90.

2nd Revision – authors' response

3 October 2018

Response to reviewer #2:

Reviewer #2 (Comments to the Authors (Required)):

The revision is improved. A few remaining issues are below.

1. p9, line 2 from the bottom, what is "(5,1% of the surface)"? and which surface?
5,1% of the combined surface of dsRBD3A and 4. The surface is the heterodimerization interface between dsRBD3A and 4. We clarified this point at p. 9, line 219.
2. p10, line 222, which figure "(Fig, B-C)"?
Thank you for noticing this. The Figure we refer to is Fig. 2. We added it in the text.
3. Many single-sentence paragraphs - p10, p13, p14, and p16 - should be avoided.
Changed.
4. p13, line 297, make sure to include the "water-mediated" in the sentence: S187 makes a similar water-mediated contact in ...

Done.

5. p13, line 307, the side chain of Q293 is directed towards the edge of C90 - please include the information about the inter-atomic distance from the current structure, and include the sentences from Response to the Reviewers' Comments in the same paragraph of the main text: the side chain of Gln293 is not in hydrogen bonding with the base pair of C90-G197, and it does not establish a clear base-specific interaction but is in the vicinity of C90.

We included the distance between Q293 and C90, which is 4.4 Å, and clearly above H-bonding distance. In Fig. 4B-E, H-bonds are conventionally marked with a dashed line while the interaction between Q293 and C90 is not. From the inter-atomic distance (now included) and from the close ups of interactions in Fig.4 it is clear that Q293 and C90 are not in H-bonding interaction. Therefore, we feel that it is not necessary to state that this interaction is NOT an H-bond in the main text. We find also that “directed towards” is a milder description of the interaction than “in the vicinity”.

3rd Editorial Decision

5 October 2018

October 5, 2018

RE: Life Science Alliance Manuscript #LSA-2018-00187-TRR

Dr. Fulvia Bono
University of Exeter
Living Systems Institute
Stocker Road
Exeter EX4 4QD
United Kingdom

Dear Dr. Bono,

Thank you for submitting your Research Article entitled "Crystal structure of Staufen1 with a physiological RNA sheds light on substrate selectivity". I appreciate the introduced changes, and it is a pleasure to let you know that your manuscript is now accepted for publication in Life Science Alliance. Congratulations on this interesting work.

The final published version of your manuscript will be deposited by us to PubMed Central (PMC) as soon as we are allowed to do so, the application for PMC indexing has been filed. You may be eligible to also deposit your Life Science Alliance article in PMC or PMC Europe yourself, which will then allow others to find out about your work by Pubmed searches right away. Such author-initiated deposition is possible/mandated for work funded by eg NIH, HHMI, ERC, MRC, Cancer Research UK, Telethon, EMBL.

Please also see:

<https://www.ncbi.nlm.nih.gov/pmc/about/authorms/>

<https://europepmc.org/Help#howsubsmenu>

DISTRIBUTION OF MATERIALS:

Again, congratulations on a very nice paper. I hope you found the review process to be constructive and are pleased with how the manuscript was handled editorially. We look forward to future exciting submissions from your lab.

Sincerely,

Andrea Leibfried, PhD
Executive Editor
Life Science Alliance
Meyrhofstr. 1
69117 Heidelberg, Germany
t +49 6221 8891 502
e a.leibfried@life-science-alliance.org
www.life-science-alliance.org